# Economic analysis of open versus laparoscopic versus robot-assisted versus transanal total mesorectal excision in rectal cancer patients: A systematic review

Ritchie T. J. Geitenbeek[1,2]*, Thijs A. Burghgraef[1,2], Mark Broekman[2], Bram P. A. Schop[2], Tom G. F. Lieverse[2], Roel Hompes[3], Klaas Havenga[2], Maarten J. Postma[4,5], Esther C. J. Consten[1,2], on behalf of the MIRECA study group[¶]

1 Department of Surgery, Meander Medical Center, Amersfoort, The Netherlands, 2 Department of Surgery, Groningen University Medical Center, University of Groningen, Groningen, The Netherlands, 3 Department of Surgery, Amsterdam University Medical Centre, Location Amsterdam Medical Centre, Amsterdam, The Netherlands, 4 Department of Health Sciences, University Medical Centre Groningen, University of Groningen, Groningen, The Netherlands, 5 Department of Economics, Econometrics & Finance, Faculty of Economics & Business, University of Groningen, Groningen, The Netherlands

¶ Collaborators of the MIRECA (Minimally Invasive RECtal CArcinoma surgery) study group are listed in the Acknowledgments.

* rtj.geitenbeek@meandermc.nl

## Abstract

### Objectives

Minimally invasive total mesorectal excision is increasingly being used as an alternative to open surgery in the treatment of patients with rectal cancer. This systematic review aimed to compare the total, operative and hospitalization costs of open, laparoscopic, robot-assisted and transanal total mesorectal excision.

### Methods

This systematic review was performed according to the Preferred Reporting Items for Systematic Reviews and Meta-Analyses statement (PRISMA) (S1 File) A literature review was conducted (end-of-search date: January 1, 2023) and quality assessment performed using the Consensus Health Economic Criteria.

### Results

12 studies were included, reporting on 2542 patients (226 open, 1192 laparoscopic, 998 robot-assisted and 126 transanal total mesorectal excision). Total costs of minimally invasive total mesorectal excision were higher compared to the open technique in the majority of included studies. For robot-assisted total mesorectal excision, higher operative costs and lower hospitalization costs were reported compared to the open and laparoscopic technique. A meta-analysis could not be performed due to low study quality and a high level of heterogeneity. Heterogeneity was caused by differences in the learning curve and statistical methods used.

**Data Availability Statement:** As this is a systematic review, all individual data is already open to the public. The included papers are mentioned in the manuscript including references.

**Funding:** The authors received no specific funding for this work.

**Competing interests:** EC is proctor for Intuitive Surgical and received a grant from this organization for the Vantage trial. No (financial) support from this organization has been received for the submitted manuscript. Neither has there been any other activities or relations that could appear to have influenced the submitted work. This does not alter our adherence to the PLOS ONE policies on sharing data and materials. All other authors declare: no support from any organization for the submitted work; no financial relationships with organizations that may have an interest in the submitted work in the previous 3 years; and no other relationships or activities that could appear to have influenced the submitted work.

## Conclusion

Literature regarding costs of total mesorectal excision techniques is limited in quality and number. Available evidence suggests minimally invasive techniques may be more expensive compared to open total mesorectal excision. High-quality economical evaluations, accounting for the learning curve, are needed to properly assess costs of the different techniques.

## Introduction

Over the past decades, significant improvements in minimally invasive approaches led to the subsequent introduction of laparoscopic, robot-assisted and transanal colorectal procedures. Laparoscopic total mesorectal excision (L-TME), robot-assisted total mesorectal excision (R-TME) and transanal total mesorectal excision (TaTME) are now more widely used, especially after their safety and feasibility was established for the surgical treatment of extraperitoneal rectal carcinoma [1–9].

A recently published multicenter randomized controlled trial by the REAL study group reported robotic surgery resulted in better oncological quality of resection and postoperative recovery than conventional laparoscopic surgery [10]. Similarly, a systematic review and meta-analysis by Milone et al. reported R-TME was associated with significantly higher rates of complete TME resection, whereas L-TME had a higher incidence of nearly complete TME [11]. Nevertheless, as the majority of previously published studies described no clear differences regarding surgical and oncological outcomes between the minimally invasive techniques, when performed by experienced surgeons, all three techniques are currently being performed as standard care [3–7].

Spiraling healthcare costs have led to an emphasis on value-based healthcare. Consequently, cost-effectiveness and 'added' value of new technologies such as minimally invasive TME must be compared to conventional techniques. It is suggested that R-TME has higher costs compared to L-TME due to increased implementation costs, costs of re-usables and longer operating times [12, 13]. However, these studies fail to account for the learning curve and do not include potential differences in postoperative outcomes of R-TME. Contrastingly, costs of TaTME were suggested to be lower compared to L-TME due to shorter operating times when using a two-team approach [3, 7, 14, 15]. However, reduced operative costs resulting from shorter operative time may be offset by higher costs of the surgical team when performing a two-team approach [16]. Finally, although a systematic review and meta-analysis is available [17], this was rendered suboptimal for cost-analyses as costs were not the primary outcome and studies using retrospective data were included. Therefore, this systematic literature review aimed to compare costs of the different TME techniques.

## Methods

This systematic review was performed according to the Preferred Reporting Items for Systematic Reviews and Meta-Analysis (PRISMA) statement and in line with the protocol agreed by all authors [18]. The study protocol was written in accordance with the PRISMA-P statement and registered in the International prospective register of systematic reviews (Prospero).

The study selection criteria were defined by applying the Population/Participants, Intervention, Comparison, Outcome, and Study design (PICOS) framework. Population: patients with

rectal cancer undergoing TME. Interventions: open TME, L-TME, R-TME and TaTME. Comparisons: studies were deemed eligible independently of head-to-head comparisons (i.e., L-TME vs. R-TME). Outcomes: operative costs, defined as all surgical costs, including costs of the operating theatre, personnel costs, costs of disposables and costs for system purchase, maintenance, and updates. Hospitalization costs, defined as all non-surgical costs, including costs of hospital ward admission, therapy, prescribed drugs, laboratory tests, consumables, housing, and personnel costs. Total costs, defined as all surgical and non-surgical costs, in general these consisted of the sum of the operative costs and hospitalization costs. However, due to variation in cost components used per study as well as definitions used, costs were reported as described in individual studies and different reporting outcomes were evaluated for inclusion on a case-by-case basis. Notably, studies reporting total hospitalization costs generally referred to these costs as the combined costs of operative and hospitalization costs. RCTs, prospective studies and retrospective studies reporting from a prospective database were included. Excluded were studies that met at least one of the following criteria: (a) not published in English, French, German, or Spanish, (b) case reports, letters, conference abstracts, commentaries, and reviews, and (c) full text unavailable. In case of overlapping populations between studies, the study with the largest sample size and longest follow-up was included.

The systematic search was supported by an experienced librarian and performed for the period of January 1st, 2000, until January 1st, 2023. Search syntaxes are provided in S2 File. Studies were assessed for eligibility through search of the PubMed, Medline, Embase, Scopus, Web of Science and Cochrane Library databases. Four independent reviewers (RG, MB, BS, TL) identified potentially eligible studies through title and abstract screening using Rayyan QCRI, a web-based software management program. Disagreements were solved through discussion, in which two additional authors were involved (MP and EC). For literature saturation, reference lists of included studies were hand-searched for potentially relevant, missing articles using systematic "snowball" procedure guidelines [19].

### Data extraction

Four independent reviewers (RG, MB, BS, and TL) extracted data of the included studies using a standardized data extraction form. Prior to data extraction training was provided for using the form. The tabulated data was used for evidence synthesis and quality assessment. Disagreements were solved through discussion, in which two additional authors were involved (MP and EC). The following data was extracted for all eligible studies: study characteristics (first author, journal and year of publication, country, study design, in/exclusion criteria, study period, type of surgery, number of patients, length of follow-up, funding, conflicts of interest and limitations), patient demographics (gender, age, technique used), preoperative data (disease stage, neoadjuvant treatment, American Society of Anaesthesiologists (ASA) score), intraoperative data (number of surgeons performing procedure, surgeon experience, operation time, type of procedure, anastomosis rate, conversion rate), postoperative data (length of ward stay, length of intensive care unit (ICU) stay, Clavien-Dindo classification of postoperative complications and mortality), and economic costs (operative costs, hospitalization costs, total costs, perspective, discount rate and model assumptions).

To compare reported costs, all costs were converted from the date of publication to 2021 using the inflation of the country where the research was conducted. These costs were then converted to US dollars by taking the purchasing power parities into account (S3 File) In case of missing relevant data, the corresponding authors of the study were requested for additional data. A reminder email was sent up to three times.

### Risk of bias and quality assessment

The quality of included studies was independently appraised by four reviewers (RG, MB, BS and TL) using the adjusted Consensus Health Economic Criteria (CHEC) tool [20] (S4 and S5 Files). The adjusted CHEC tool consists of 18 questions, resulting in a score ranging between 0–18 with a score of ≥12 denoting high quality. In the item assessing whether an appropriate time horizon was used for relevant outcomes to occur, the cut-off was a priori set at 3 months postoperatively. Regarding the item assessing if relevant costs were given in relation to the perspective, for studies reporting long term costs these included hospitalization costs, operative costs, post-discharge costs and indirect costs, while for studies reporting only hospitalization costs these included hospitalization costs and operative costs. Disagreements were resolved through discussion, in which two additional authors were involved (MP and EC).

### Data analysis

A narrative synthesis was performed to present the findings of the included studies following the European Social Research Council Guidance on the Conduct of Narrative Synthesis in Systematic Reviews [21]. Categorical data was summarized using numbers and percentages. Continuous variables were summarized as means and standard deviations (SDs) or median and interquartile ranges (IQRs). Operative, hospitalization and total costs were compared between techniques. Studies comparing techniques head-to-head were prioritized. A meta-analysis will be performed if studies are methodologically and statistically homogeneous.

## Results

### Study selection

After removal of duplicates, a total of 9780 citations were identified (Fig 1). Following title and abstract screening, 93 studies were selected for full-text evaluation. Ultimately, 12 studies met the predetermined search criteria and were selected for qualitative synthesis [10, 22–32]. A meta-analysis was not performed, due to heterogeneity resulting from differences in the learning curve of the performing surgeon and heterogeneity among the included cost components. Furthermore, heterogeneity resulted from differences in primary effect measures and statistical methods used across the small number of included studies, lack of standard deviations or other statistical measures.

### Study characteristics

Five studies compared L-TME vs. R-TME(10,25,27,28,31), one L-TME vs. open TME [22], one L-TME vs. TaTME [29], and one L-TME vs. R-TME vs. open TME [23]. No studies comparing R-TME with TaTME, or open TME with R-TME were included in this review. Additionally, four studies only reported costs of a single technique [25, 28, 31, 32]. A total of 2542 rectal cancer patients were included, 1192 (46.9%) of whom underwent L-TME, 998 (39.3%) R-TME, 226 (8.9%) open TME and 126 (5.0%) TaTME. Demographics of the included studies are provided in Table 1. Patient demographics, preoperative characteristics, operative data and postoperative outcomes are provided in S1 and S2 Tables. Cost components and total costs of the included studies are provided in Table 2. All R-TMEs were performed using the da Vinci surgical system.

### Quality assessment of included studies

The mean CHEC tool score was 6.1 ± 1.6, which indicates a low methodological quality of the included studies. There were no studies of high quality (score of ≥12). Reasons for low score

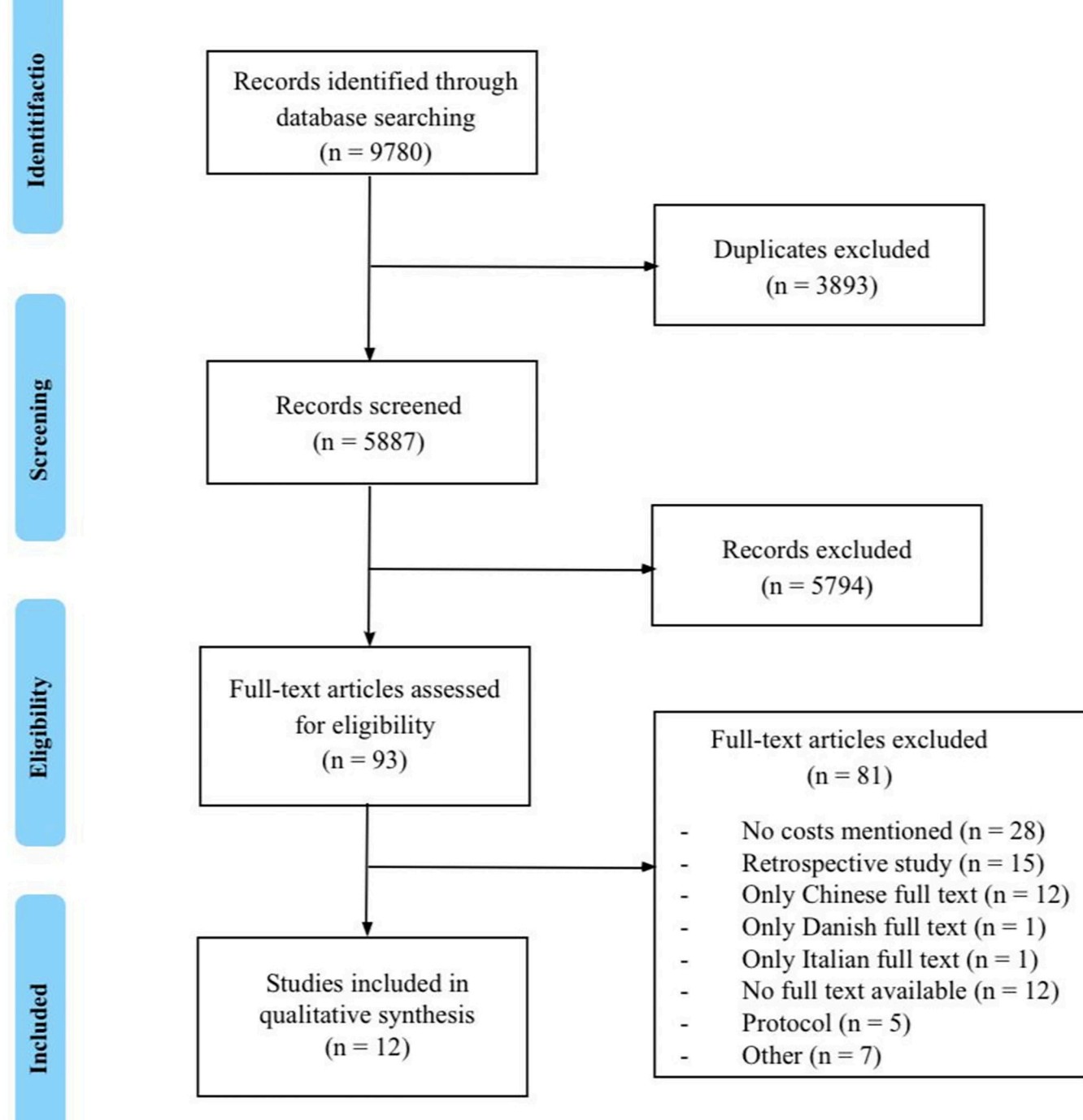

**Fig 1. Flowchart of included studies.** *n*; number.

**Table 1. Demographics of the studies included in this review.**

| First author | Year | Country | Study design | Method | Study period | Follow-up | # Total | # O-TME | # L-TME | # R-TME | #TaTME |
|---|---|---|---|---|---|---|---|---|---|---|---|
| Baek et al | 2010 | USA/ South Korea | NRRS | - | 04/2003–03/2009 | 30 days | 82 | NA | 41 | 41 | NA |
| Candido et al | 2020 | Italy | NRRS | Cost analysis | L-TME: 02/2014–10/2018 TaTME: 01/2015–10/2018 | 30 days | 152 | NA | 66 | NA | 86 |
| Elbarmelgi et al | 2022 | Egypt | NRPS | - | 05/2018–01/2020 | 6 months | 40 | NA | NA | NA | 40 |
| Feng et al | 2014 | China | RCT | - | 11/2011–11/2012 | 30 days | 59 | NA | 59 | NA | NA |
| Feng et al | 2022 | China | RCT | - | 07/2016–12/2020 | 30 days | 1171 | NA | 585 | 586 | NA |
| Leung et al | 2004 | China | RCT | - | 09/1993–10/2002 | 5 years | 403 | 200 | 203 | NA | NA |
| Morelli et al | 2018 | Italy | NRRS | Cost analysis | 04/2010–04/2017 | - | 80 | NA | NA | 80 | NA |
| Pai et al | 2015 | USA | NRRS | - | 08/2005–10/2012 | - | 24 | NA | NA | 24 | NA |
| Pan et al | 2022 | China | NRRS | - | 01/2019–01/2021 | 5 years | 106 | NA | 56 | 50 | NA |
| Park et al | 2015 | South Korea | NRPS | - | 04/2006–08/2011 | 2 years | 217 | NA | 84 | 133 | NA |
| Ramji et al | 2015 | Canada | NRRS | Cost analysis | 01/2007–12/2013 | 5 years | 79 | 26 | 27 | 26 | NA |
| Rouanet et al | 2020 | France | NRRS | Cost analysis | 01/2011–12/2018 | - | 129 | NA | 71 | 58 | NA |
| Total | | | | | | | 2542 | 226 | 1192 | 998 | 126 |

*L-TME*, laparoscopic total mesorectal excision; *NA*, not applicable; *NRPS*, non-randomized prospective study; *NRRS*, non-randomized retrospective study reporting from prospective database; *O-TME*, open total mesorectal excision; *R-TME*, robotic total mesorectal excision; *RCT*, randomized controlled trial; *TaTME*, transanal total mesorectal excision; *#*, number; *-*, not available

were mainly studies not being economic evaluations, inadequate study design for cost assessment and incomplete description of cost components.

## Results of studies comparing techniques head-to-head

Eight studies compared costs of two or more techniques[10, 22–24, 26, 27, 29, 30]. These studies included a total of 1133 L-TME, 894 R-TME, 226 open TME and 86 TaTME patients.

**Laparoscopic vs. open TME.** One study (Leung et al.) [22] compared costs of L-TME and open TME. This study reported significantly higher total costs of L-TME compared to open TME ($13,336.21 vs. $10,879.02; p<0.001).

**Laparoscopic vs. transanal TME.** One study (Candido et al.) [29] compared costs of L-TME and TaTME. This study reported higher total costs of TaTME compared to L-TME ($2,809.72 vs $1,634.37). Operative costs of TaTME were $1,175.35 higher compared to L-TME, while hospitalization costs were comparable.

**Laparoscopic vs. robot assisted TME.** Five studies compared costs of L-TME and R-TME [10, 24, 26, 27, 30]. Four studies (Feng et al, Pan et al, Park et al, Rouanet et al.) [10, 26, 27, 30] reported significantly higher total costs of R-TME compared to L-TME ($11,733.61 vs $7,436.97; p<0.0001, $12,544.63 vs $8,746.01; p<0.001, $14,567.89 vs $11,548.33; p = 0.005 and $18,929.71 vs $15,662.73, respectively), while one study (Baek et al.) [24] reported no significant differences ($104,278.08 vs $77,791.96; p = 0.092). Two studies (Feng et al, Rouanet et al.) [10, 27] reported higher operative costs for R-TME ($8,471.48 vs $3,883.36; p<0.001, Feng et al), while the hospitalization costs were lower for R-TME ($3,131.11 vs $3,492.27; p = 0.002, $5,886.84 vs $6,942.52; p = 0.008, respectively). In the study by Rouanet et al, this was partly explained by higher instrumentation costs. Instrumentation costs were higher in

**Table 2. Overview of total costs, operative costs, hospitalization costs and reported cost components of the included studies.**

| Study | Technique | Total costs | Operative costs | | | | | | Hospitalization costs | | | | | | | |
|---|---|---|---|---|---|---|---|---|---|---|---|---|---|---|---|---|
| | | | Total | Theatre | Conversion | Instruments | Personnel | Consumables | Total | Ward | Complication | Lab | Imaging | Pharmacy | PACU | ICU |
| Baek et al | L-TME | $77,791.96 | - | - | - | - | - | - | - | - | - | - | - | - | - | - |
| | R-TME | $104,278.08 | - | - | - | - | - | - | - | - | - | - | - | - | - | - |
| Candido et al | L-TME | $1,634.37 | X | - | - | X | - | X | - | - | - | $71.65 | $61.83 | $44.34 | - | - |
| | TaTME | $2,809.72 | X | - | - | X | - | X | - | - | - | $79.12 | $61.52 | $61.37 | - | - |
| Elbarmelgi et al | TaTME | $2,788.90 | - | - | - | X | - | X | - | X | - | - | - | - | - | - |
| Feng et al | L-TME | $11,166.91 | X | - | - | - | - | - | X | X | - | X | X | X | - | - |
| Feng et al | L-TME | $7,436.97 | $3,883.46 | - | - | - | - | - | $3,429.27 | - | - | - | - | - | - | - |
| | R-TME | $11,733.61 | $8,471.48 | - | - | - | - | - | $3,131.11 | - | - | - | - | - | - | - |
| Leung et al | O-TME | $10,879.02 | - | - | - | X | - | - | - | - | - | - | - | - | - | - |
| | L-TME | $13,336.21 | - | - | - | X | - | - | - | - | - | - | - | - | - | - |
| Morelli et al | Si | $16,030.03 | - | - | - | - | $2,125.36 | $6,856.23 | $7,013.37 | - | - | - | - | - | - | - |
| | Xi | $12,030.35 | - | - | - | - | $1,870.41 | $3,546.00 | $5,167.84 | - | - | - | - | - | - | - |
| Pai et al | R-TME | $25,883.23 | - | - | - | - | - | - | - | - | - | - | - | - | - | - |
| Pan et al | L-TME | $8,746.01 | - | - | - | - | - | - | X | - | - | - | - | - | - | - |
| | R-TME | $12,544.63 | - | - | - | - | - | - | X | - | - | - | - | - | - | - |
| Park et al | L-TME | $11,548.33 | X | - | - | - | - | - | - | - | - | - | - | - | - | - |
| | R-TME | $14,567.89 | X | - | - | - | - | - | - | - | - | - | - | - | - | - |
| Ramji et al | O-TME | $11,213.94 | $3,874.99 | - | - | - | - | - | - | $3,922.54 | - | $907.18 | $38.04 | $447.95 | $553.31 | - |
| | L-TME | $10,262.96 | $4,744.67 | - | - | - | - | - | - | $4,388.16 | - | $921.65 | $65.31 | $516.68 | $414.74 | - |
| | R-TME | $16,316.85 | $10,607.72 | - | - | - | - | - | - | $3,934.29 | - | $506.88 | $70.20 | $385.37 | $521.41 | - |
| Rouanet et al | L-TME | $15,662.73 | - | $2,185.65 | $511.71 | $2,279.58 | - | - | $6,942.52 | - | $2,743.64 | - | - | - | - | $999.60 |
| | R-TME | $18,929.71 | - | $2,416.97 | $107.94 | $4,717.60 | - | - | $5,886.84 | - | $4,605.44 | - | - | - | - | $1,219.71 |

*ICU*, intensive care unit; *L-TME*, laparoscopic total mesorectal excision; *O-TME*, open total mesorectal excision; *PACU*, post-anesthesia care unit; *R-TME*, robotic total mesorectal excision; *TaTME*, transanal total mesorectal excision; *X*, study reported including cost component though costs not provided in study; *-*, cost component was not reported

R-TME than L-TME ($4,717.60 vs $2,279.58), while conversion costs were lower in R-TME than L-TME ($107.94 vs $511.71). Lastly, costs of complications and ICU stay were significantly higher in R-TME than L-TME ($4,605.44 vs $2,743.64, and $1,219.71 vs $999.60; p<0.001, respectively).

**Laparoscopic vs. robot assisted vs. open TME.** One study (Ramji et al.) [23] reported costs of L-TME, R-TME and open TME. This study reported significantly higher total costs of R-TME compared to L-TME and open TME ($16,316.85 vs $10,262.96 vs $11,213.94; p = 0.029). Operative costs were significantly higher in R-TME compared to L-TME and open ($10,607.72 vs $4,744.67 vs $3,874.99; p<0.001). Hospitalization costs showed laboratory costs were significantly lower in R-TME than L-TME and open TME ($506.88 vs $921.65 vs $907.18; p<0.001). PACU costs were significantly lower in L-TME compared to R-TME and open TME ($414.74 vs $521.41 vs $553.31; p = 0.011). No significant differences between the three techniques were found for ward, imaging and pharmacy costs.

### Results of studies assessing one technique

Four studies reported costs of one technique [25, 28, 31, 32]. These studies included a total of 59 L-TME, 104 R-TME, 40 TaTME patients. One study (Feng et al.) [25] reported costs of L-TME. This study reported total costs of $11,166.91. One study (Elbarmelgi et al.) [32] reported costs of TaTME. This study reported total costs of $2,788.90. Two studies (Pai, Morelli et al. [28, 31]) reported costs of R-TME. One study (Pai et al.) [28] reported total costs of $25,883.23. One study (Morelli et al.) [31] reported significantly lower total costs of Xi-R-TME compared to Si-R-TME ($12,030.35 vs $16,030.03; p<0.001). Lower total costs resulted from significantly lower operative costs in Xi-R-TME ($1,756.11 vs $2,424.23; p<0.001) as well as significantly lower hospitalization costs consisting of non-theatre costs and consumable costs ($5,167.84 vs $7,013.37 and $5,422.01 vs $6,856.23; p<0.001, respectively).

## Discussion

This systematic review aimed to provide an overview of the current literature regarding costs of open TME, L-TME, R-TME and TaTME, and reveals a lack of high-quality studies. A limited number of available studies suggest that total costs of the minimally invasive techniques are higher compared to their open counterpart. This may be explained by higher operative costs of the minimally invasive techniques. In terms of hospitalization costs, R-TME may be less expensive than L-TME. However, there was large heterogeneity between studies and available studies were at high risk of bias due to inadequacy in accounting for different cost components.

Only two studies compared minimally invasive TME to open TME [22, 23]. The reported total costs and operative costs of the minimally invasive techniques were higher compared to open TME. This is in line with other retrospective studies evaluating costs of minimally invasive and open TME [33, 34]. Within these studies, the higher total costs were attributed higher operative costs, which are only partially mitigated by lower hospitalization costs, resulting from a decrease in complications and length of hospital stay. Notably, whereas the aforementioned studies reported higher total costs of the minimally invasive techniques, King et al. [35] conflictingly reported lower total costs. Notably, this study was performed from the perspective of the UK National Health Service and included indirect costs (number of days that patients in paid work (full or part time) took off for their condition) until three months after surgery. Accounting for improved postoperative outcomes inherent to minimally invasive surgery, the authors described significantly lower indirect costs and consequently total costs of L-TME compared to open TME. This may suggest current literature reporting total costs

based on only direct in-hospital costs may overestimate the cost difference between minimally invasive techniques and open TME.

Additionally, the hospitalization costs and operative costs of R-TME and TaTME may be higher compared to L-TME. Only one study compared TaTME with L-TME and reported significantly higher total costs of TaTME [29]. No differences in hospitalization costs were reported, suggesting higher operative costs contributed to the higher total costs. As reported by Vignali et al., currently available evidence suggests short-term (post)operative and oncologic results of TaTME are comparable to L-TME [36]. However, analyses of the costs related to the procedure are required, in particular when a two-team operation is performed. Importantly, all TaTME procedures in the study by Candido et al. were performed using a two-team approach, which is known to significantly reduce operating times and consequently operative costs [26]. However, it is unclear whether the authors accounted for these additional surgical personnel costs. If these costs have yet to be accounted for, these might offset the initially lower operative costs due to the shorter operative times. Regarding the costs of R-TME, six studies [10, 23, 24, 26, 27, 30] reported significantly higher total costs compared to L-TME. These findings are in line with other retrospective studies reporting higher total costs of R-TME [13, 37, 38]. Conflicting outcomes were found between the three studies reporting both operative costs and hospitalization costs [10, 23, 27]. Ramji et al. reported higher operative costs and equal hospitalization costs, whereas Feng et al. and Rouanet et al. reported higher operative costs and lower hospitalization costs. Contrasting to current literature, Rouanet et al reported higher complication and ICU costs [37, 38]. This may be explained as this study assessed patients receiving R-TME in 2015 performed by surgeons of which some may still have been in their respective robotic learning curve, which is associated with increased morbidity [23]. Moreover, when assessing a cohort of patients receiving R-TME in 2018, after the robotic learning curve had been achieved and the team was familiar with the ERAS procedures involved, the authors reported that costs were reduced to a level that was comparable to L-TME, with lower costs in patients without co-morbidity. The total costs reported by Baek et al. [24] were strikingly high compared to other studies, which likely resulted from differences in countries' healthcare system. Perhaps the higher operative costs of TaTME and R-TME are attributed to fewer reusable instruments being available compared to L-TME. However, with more surgical systems and reusable instruments from other medical companies entering the market (i.e., HUGO RAS, Senhance Surgical System, CMR surgical system) [39, 40], instrument costs for these new techniques will likely decrease. Along this line of thought, it is noted that all studies reporting higher operative costs of R-TME used the Da Vinci Si, whereas the newer model Xi may be associated with lower operative times as redocking is not required [31]. Therefore, the total costs of R-TME may be overestimated in current literature and is likely to be lower in future studies reporting on the da Vinci Xi.

The statements regarding potentially higher total costs of minimally invasive techniques compared to open TME should be interpreted with caution. Currently, high-quality evidence is lacking, with few comparative studies available, and large heterogeneity between studies. This is mainly caused by differences in the learning curve and statistical methods used. Firstly, heterogeneity due to differences in the learning curve among studies exists: while some studies report on early experience, others studies report on surgeons far beyond their learning curves. As differences between surgeons' experience have previously been known to confound the assessment of minimally invasive TME outcomes [41], it could be doubted whether comparing techniques performed throughout different stages of the learning curve is suitable for economic evaluation. As the learning curve is known to be associated with longer operative times [42] and added morbidity which likely confounds the reported operative costs [23]. Particularly as most of the available studies reporting on minimally invasive techniques were early

experience reports with surgeons still in their learning curves. Furthermore, as more colorectal surgeons adopt minimally invasive TME, a gradual increase in surgical skills and expertise is to be anticipated [15, 43]. Consequently, morbidity rates and hospital stay duration will likely decrease in favor these new techniques, lowering hospitalization costs [31].

Secondly, regarding heterogeneity among studies caused by the used statistical analyses, differences could be due to the cost components used to measure costs, the used statistical method and the used follow-up period. Regarding the cost components used to measure costs, the majority of studies only included in-hospital operative costs in the cost comparison. Hence, it was expected that minimally invasive techniques would be associated with higher operative costs compared to open surgery, as the instruments and console are significant cost items, while operative times are longer [31]. However, operative costs only reflect part of the total costs. Instead, a societal perspective including indirect costs (i.e. loss of productivity, time spent on family caregiving) should be used to illuminate potential differences between technique costs. Perhaps, including indirect costs will favour minimally invasive techniques offering improved outcomes. This might be of importance as R-TME and TaTME have been reported to achieve improved postoperative outcomes compared to other techniques, hence the direct costs described in individual studies may belittle the true economic benefits of these techniques [10, 43]. Furthermore, as the included studies were not economic analyses, they failed to comply with the methodological items recommended by the CHEC checklist. Consequently, no clear description was provided of the cost components used to calculate different costs. These cost components not included in the evaluation may have led to wrong conclusions. For instance, the total costs of R-TME were significantly higher compared to other techniques, even while the majority of the included studies did not account for the costs of purchase and maintenance of the robot, although they represent important opportunity costs. In line with this, in order to provide comparable outcomes, clear definitions regarding included cost components should be used. Though purchase costs of the robot decreased over the past decade, they represent considerable fixed costs ($1.000.000) along with maintenance costs which vary between $90.000–175.000 per year [44]. Therefore, it is expected that the costs of R-TME may have been underestimated in these studies. Likewise, purchasing and maintenance costs of the laparoscopic platform were not accounted for in studies assessing the costs of L-TME. As the average life-span of the robot is ten years, these costs have to be addressed in the light of the number of robot-assisted surgeries performed in a center [45]. With high-volume centers performing over 400 robot-assisted surgeries annually and low-volume centers performing as low as 50, R-TME may be only cost-effective in high-volume centers. Hence, in future economic analyses researchers should take into account the costs of purchase and maintenance as well as center volume. Furthermore, staff salaries were not included as transparent cost estimation and studies calculated operative costs as hourly wages times the length of surgery. As hourly wages might indicate average costs of the whole surgical team, they fail to address differences between individual surgeons. In addition, costs were based on hospital charges, which represent local estimations of costs related to specific procedures and may vary widely between different countries and reimbursement systems. This systematic review has strengths including: (a) thorough comprehensive search of the literature performed with a professional librarian, (b) using standardized pre-piloted forms for meticulous data extraction by four independent researchers, (c) scrupulous quality of evidence assessment of the studies included, and (d) recalculation of all cost outcomes via the inflation rate per county in 2021 and via purchasing power parity. Nonetheless, this systematic review cannot draw a definite conclusion regarding cost differences between the different techniques. Clearly, more high-quality studies are necessary to illuminate cost differences between techniques and enable cost considerations to be considered and allow for value-based healthcare.

We suggest that this should preferably be performed with large, randomized trials or high quality non-randomized prospective studies, while controlling for differences in the learning curve. In addition, these comparative studies should be performed from a societal perspective and include indirect costs, particularly as these are not yet included in currently available evidence [16, 22, 23]. Furthermore, we propose that trials adhering to these recommendations, such as the VANTAGE trial [46], investigate whether increasing experience with minimally invasive techniques, decreasing costs of instrumentation, and potentially improved postoperative outcomes offset the initially higher costs of these techniques.

## Supporting information

**S1 File. PRISMA checklist.**
(PDF)

**S2 File. Search syntax.**
(PDF)

**S3 File. Overview of original costs and converted costs.**
(PDF)

**S4 File. Modified Consensus on Health Economic Criteria (CHEC) checklist.** Modified Consensus on Health Economic Criteria (CHEC) checklist tool. Questions used per category and explanatory note are provided.
(PDF)

**S5 File. Quality assessment of the included studies using the Consensus on Health Economic Criteria (CHEC) checklist.** NA, not applicable. The Consensus on Health Economic Criteria (CHEC) checklist is a current research standard for conducting systematic reviews which are based on economic evaluation studies and was used to rate study the quality of all studies included in this review. The results of the CHEC tool indicate risk of bias and applicability concerns. This study used a modified CHEC tool using categorical questions adjusted to study design of the included studies.
(PDF)

**S1 Table. Patients demographics and preoperative characteristics of included studies.** ASA, American Society of Anesthesiology; BMI, body mass index; L-TME, laparoscopic total mesorectal excision; n, number of patients; O-TME, open total mesorectal excision; R-TME, robotic total mesorectal excision; TaTME, transanal total mesorectal excision; -, not available.
(PDF)

**S2 Table. Operative data and postoperative data.** APR, abdominoperineal resection; CRM+, positive circumferential resection margin; L-TME, laparoscopic total mesorectal excision; LAR, low anterior resection; n, number of patients; NA, not applicable; O-TME, open total mesorectal excision; R-TME, robotic total mesorectal excision; TaTME, transanal total mesorectal excision; -, not available, * Data shows median values, ᵃ Other procedures include coloanal anastomosis, intersphincteric resection and Hartmann surgery.
(PDF)

## Acknowledgments

We would kindly like to thank all collaborators of the MIRECA (Minimally Invasive RECtal CArcinoma surgery) study group for their participation in the development of this study

project and protocol and K. Sijtsma of University Medical Center Groningen for her support with the development of the search strategy.

MIRECA study group: G.J.D. van Acker, T.S. Aukema, H.J. Belgers, F.H. Beverdam, J.G. Bloemen, K. Bosscha, S.O. Breukink, P.P.L.O. Coene, R.M.P.H. Crolla, P. van Duijvendijk, E. B. van Duyn, I.F. Faneyte, S.A.F. Fransen, A.A.W. van Geloven, M.F. Gerhards, W.M.U. van Grevenstein, K. Havenga, I.H.J.T. de Hingh, C. Hoff, G. Kats, J.W.A. Leijtens, M.F. Lutke Hol-zik, J. Melenhorst, M.M. Poelman, A. Pronk, A.H.W. Schiphorst, J.M.J. Schreinemakers, C. Sietses, A.B. Smits, I Somers, E.J. Spillenaar Bilgen, H.B.A.C. Stockmann, A.K. Talsma, P.J. Tanis, J. Tuynman, E.G.G. Verdaasdonk, F.A.R.M. Warmerdam, H.L. van Westreenen, D.D.E. Zimmerman.

## Author Contributions

**Conceptualization:** Ritchie T. J. Geitenbeek, Thijs A. Burghgraef, Mark Broekman, Bram P. A. Schop, Tom G. F. Lieverse, Roel Hompes, Klaas Havenga, Maarten J. Postma, Esther C. J. Consten.

**Data curation:** Ritchie T. J. Geitenbeek, Thijs A. Burghgraef, Mark Broekman, Bram P. A. Schop, Tom G. F. Lieverse.

**Formal analysis:** Ritchie T. J. Geitenbeek, Thijs A. Burghgraef, Mark Broekman, Bram P. A. Schop, Tom G. F. Lieverse.

**Methodology:** Thijs A. Burghgraef, Roel Hompes, Klaas Havenga, Maarten J. Postma, Esther C. J. Consten.

**Project administration:** Ritchie T. J. Geitenbeek, Mark Broekman, Bram P. A. Schop, Tom G. F. Lieverse.

**Supervision:** Roel Hompes, Klaas Havenga, Maarten J. Postma, Esther C. J. Consten.

**Validation:** Roel Hompes, Klaas Havenga, Maarten J. Postma, Esther C. J. Consten.

**Writing – original draft:** Ritchie T. J. Geitenbeek, Mark Broekman, Bram P. A. Schop, Tom G. F. Lieverse.

**Writing – review & editing:** Thijs A. Burghgraef, Roel Hompes, Klaas Havenga, Maarten J. Postma, Esther C. J. Consten.

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
