## [Decision Letter · Decision Letter 0]

24 Apr 2023

PONE-D-23-09962Economic analysis of open versus laparoscopic versus robot-assisted versus transanal total mesorectal excision in rectal cancer patients: a systematic review.PLOS ONE

Dear Dr. Geitenbeek,

Thank you for submitting your manuscript to PLOS ONE. After careful consideration, we feel that it has merit but does not fully meet PLOS ONE’s publication criteria as it currently stands. Therefore, we invite you to submit a revised version of the manuscript that addresses the points raised during the review process.

We look forward to receiving your revised manuscript.

Kind regards,

Michele Manigrasso

Academic Editor

PLOS ONE

Journal Requirements:

"This research received no specific grant from any funding agency in the public, commercial or not-for-profit sectors. "

"EC is proctor for Intuitive Surgical and received a grant from this organization for the Vantage trial. No (financial) support from this organization has been received for the submitted manuscript. Neither has there been any other activities or relations that could appear to have influenced the submitted work. All other authors declare: no support from any organization for the submitted work; no financial relationships with organizations that may have an interest in the submitted work in the previous 3 years; and no other relationships or activities that could appear to have influenced the submitted work. "

Additional Editor Comments:

Dear Author,

thank you for submitting the manuscript to Plos One.

I think that the manuscript is very interesting, but withouth enough priority for publication in its present form.

Please see and respond to the reviewer's comment.

Reviewers' comments:

Reviewer's Responses to Questions

**Comments to the Author**

1. Is the manuscript technically sound, and do the data support the conclusions?

Reviewer #1: Yes

2. Has the statistical analysis been performed appropriately and rigorously? 

Reviewer #1: Yes

3. Have the authors made all data underlying the findings in their manuscript fully available?

Reviewer #1: Yes

4. Is the manuscript presented in an intelligible fashion and written in standard English?

Reviewer #1: Yes

5. Review Comments to the Author

Reviewer #1: The study is well structured but there are some points to clarify:

- In the section “laparoscopic versus open TME” Leung et al. reported significantly higher total costs of L-TME compared to open TME. Is there any information about specific fields such as hospitalization costs, post-discharge costs and indirect costs in current literature?

- In the section “laparoscopic versus transanal TME” Candido et al. reported higher total costs of TaTME compared to L-TME and in details operative costs of TaTME were higher compared to L-TME, while hospitalization costs were comparable. Is there any information about specific fields such as instrumentation costs, conversion costs, post-discharge costs and indirect costs in current literature?

Moreover evaluating the comparison between laparoscopic and transanal TME, why don’t you mention this paper?

Vignali A, Elmore U, Milone M, Rosati R. Transanal total mesorectal excision (TaTME): current status and future perspectives. Updates Surg. 2019 Mar;71(1):29-37. doi: 10.1007/s13304-019-00630-7. Epub 2019 Feb 8. PMID: 30734896.

- In the section “laparoscopic versus robot-assisted TME” it has been reported that “four studies showed significantly higher total costs of R-TME compared to L-TME while one study reported no significant differences. Two studies reported higher operative costs for R-TME while the hospitalization costs were lower for R-TME. Instrumentation costs were higher in R-TME than L-TME while conversion costs were lower in R-TME than L-TME. Lastly, costs of complications and ICU stay were significantly higher in R-TME than L-TME”.

How do you explain the higher costs of complications and ICU stay in R-TME than L-TME? This topic should be enhanced.

Moreover evaluating the comparison between laparoscopic and robot-assisted TME, why don’t you mention this paper?

Milone M, Manigrasso M, Velotti N, Torino S, Vozza A, Sarnelli G, Aprea G, Maione F, Gennarelli N, Musella M, De Palma GD. Completeness of total mesorectum excision of laparoscopic versus robotic surgery: a review with a meta-analysis. Int J Colorectal Dis. 2019 Jun;34(6):983-991. doi: 10.1007/s00384-019-03307-0. Epub 2019 May 6. PMID: 31056732.

- In the section “laparoscopic versus robotic versus open TME” Ramji et al. reported significantly higher total costs of R-TME compared to L-TME and open TME, operative costs were significantly higher in R-TME compared to L-TME and open. Is there any information about specific fields such as post-discharge costs and indirect costs in current literature?

6. PLOS authors have the option to publish the peer review history of their article (what does this mean?). If published, this will include your full peer review and any attached files.

Reviewer #1: No

---

## [Author Response · Author response to Decision Letter 0]

15 Jun 2023

Professor Emily Chenette 

Editor-in-chief, PLoS One

PLoS One Editorial Office

184 Cambridge Science Park

Milton, Cambridge CB4 0GA

United Kingdom

 Groningen, the Netherlands. June 8th 2023

Dear Professor Emily Chenette, editor-in chief of PLoS One,

We would like to thank the reviewers for their review by providing comments and feedback, hereby enhancing the manuscript: “Economic analysis of open versus laparoscopic versus robot-assisted versus transanal total mesorectal excision in rectal cancer patients: a systematic review.” We would hereby like to send the revised manuscript for appraisal. We have taken all of the mentioned comments into account while optimizing and changing our manuscript. 

Enclosed you will find the revised manuscript. In yellow you will find the changes made in the manuscript as a reaction on the comments. In addition, below we provided an itemized, point-by-point response to the comments of the reviewers, including the page numbers of the revised manuscript. 

We hope that you might consider the adjusted manuscript for publication. 

Yours sincerely, 

On behalf of the whole study team, 

Ritchie T. J. Geitenbeek, MD, PhD-candidate

University Medical Center Groningen, Department of Surgery

Meander Medical Center, Department of Surgery

E-mail: r.t.j.geitenbeek@umcg.nl

 

General comments:

Comment #1: 

- Please ensure that your manuscript meets PLOS ONE's style requirements, including those for file naming

We have taken great care to make sure that our manuscript and files meet the PLOS ONE’s style requirements. We have made several modifications to our manuscript by using the PLOS ONE style template. We have added the following sentence, including the indicating symbol, on the Title page on lines 24 & 25: ^ Collaborators of the MIRECA (Minimally Invasive RECtal CArcinoma surgery) study group are listed in the Acknowledgments. We have changed the following words in the “Abstract” section on page 4, line 85 from: “Supplementary File 1” into: “S1 File”. We have changed the following words in the “Methods” section on page 8, line 168 from: “Supplementary File 2” into: “S2 File”. We have changed the following words in the “Data extraction” section on page 9, line 196 from: “Supplementary File 3” into: “S3 File”. We have changed the following words in the “Risk of bias and quality assessment” section on page 9, line 200 & 203 from: “Supplementary File 4 and 5” into: “S4 and S5 Files”. We have changed the following word in the “Study selection” section on page 10, line 224 from: “Fig. 1” into: “Fig 1”. We have changed the following word in the “Study selection” section on page 11, line 233 from: “Fig. 1” into: “Fig 1”. We have changed the following words in the “Study characteristics” section on page 11, line 243 from: “Supplementary Table 1 and 2” into: “S1 and S2 Tables”. Regarding the reference citations, we have changed all parenthesis into brackets throughout our manuscript. Furthermore, we changed our File naming to meet the PLOS ONE style requirements. 

Comment #2: 

- Thank you for stating the following financial disclosure: "This research received no specific grant from any funding agency in the public, commercial or not-for-profit sectors”. If you did not receive any funding for this study, please state: “The authors received no specific funding for this work.”

We have changed the following sentence in the “Funding statement” section on page 2, line 37 from: “This research received no specific grant from any funding agency in the public, commercial or not-for-profit sectors.” into: “The authors received no specific funding for this work.”

Comment #3: 

- Thank you for stating the following in the Competing Interests section: "EC is proctor for Intuitive Surgical and received a grant from this organization for the Vantage trial. No (financial) support from this organization has been received for the submitted manuscript. Neither has there been any other activities or relations that could appear to have influenced the submitted work. All other authors declare: no support from any organization for the submitted work; no financial relationships with organizations that may have an interest in the submitted work in the previous 3 years; and no other relationships or activities that could appear to have influenced the submitted work. "Please confirm that this does not alter your adherence to all PLOS ONE policies on sharing data and materials, by including the following statement: "This does not alter our adherence to PLOS ONE policies on sharing data and materials.”

We have added the following statement to the “Competing interests” section on page 2, line 30 & 31: “This does not alter our adherence to the PLOS ONE policies on sharing data and materials.”

Comment #4: 

After thoroughly reviewing our references, we have adjusted reference number 17, as this reference had not been correctly exported from our reference manager program. Subsequently, we have checked all references, ensuring that alle are complete and correct. We have not cited any retracted articles in our paper.

Reviewer 1:

Comment #1:

- In the section “laparoscopic versus open TME” Leung et al. reported significantly higher total costs of L-TME compared to open TME. Is there any information about specific fields such as hospitalization costs, post-discharge costs and indirect costs in current literature?

Thank you kindly for reviewing our manuscript and for providing this comment. We agree with the reviewer that insight into the specific fields (such as hospitalization costs, post-discharge costs and indirect costs) of the total costs would aid in discerning the cost difference between open TME and L-TME. Regrettably, the paper by Leung et al. does not focus primarily on the total costs of the two TME techniques. Besides the total costs of the techniques, no information on the cost components or any other relevant details contributing to the total costs are reported in the paper. 

Apart from our systematic review, we identified three additional articles that report on the total costs of open TME compared to minimally invasive TME that are mentioned in the “Discussion” section on page 17, paragraph 2. We agree with the reviewer that supplementing the article with further information regarding the cost components would benefit the readership of this article. Therefore, we have added the following sentence to the “Discussion” section, on page 17, paragraph 2, line 328, 329 & 330: “Within these studies, the higher total costs were attributed to elevated operative costs, which were only partially mitigated by lower hospitalization costs, resulting from a decrease in complications and length of hospital stay.” The articles did not contain any further information regarding post-discharge costs and indirect costs. Hence, we proposed for more high quality research that also include the indirect cost in their cost analysis in lines 446, 447 & 448. Also, we have changed the following sentence in the “Discussion” section on page 22, paragraph 1, line 446, 447 & 448. from: ”In addition, these comparative studies should be performed from a societal perspective and include indirect costs.” into: In addition, these comparative studies should be performed from a societal perspective and include indirect costs, particularly as these are not yet included in currently available evidence.”

Comment #2:

- In the section “laparoscopic versus transanal TME” Candido et al. reported higher total costs of TaTME compared to L-TME and in details operative costs of TaTME were higher compared to L-TME, while hospitalization costs were comparable. Is there any information about specific fields such as instrumentation costs, conversion costs, post-discharge costs and indirect costs in current literature? Moreover evaluating the comparison between laparoscopic and transanal TME, why don’t you mention this paper? Vignali A, Elmore U, Milone M, Rosati R. Transanal total mesorectal excision (TaTME): current status and future perspectives. Updates Surg. 2019 Mar;71(1):29-37. doi: 10.1007/s13304-019-00630-7. Epub 2019 Feb 8. PMID: 30734896.

We sincerely appreciate this remark and the valuable additional literature. In response to the reviewers remark regarding specific cost field, we agree that insight into these components would add value to the comparison between L-TME and TaTME. However, despite reporting the inclusion of instrumentation into the total costs, the article did not provide the actual instrumentation costs in the article. No additional information on the cost components or any other relevant details contributing to the total costs were reported in the article, apart from those as provided in Table 2. Hence, we proposed for more high quality research that also include the indirect cost in their cost analysis in lines 446, 447 & 448. Also, we have changed the following sentence in the “Discussion” section on page 22, paragraph 1, line 446, 447 & 448 from: ”In addition, these comparative studies should be performed from a societal perspective and include indirect costs.” into: In addition, these comparative studies should be performed from a societal perspective and include indirect costs, particularly as these are not yet included in currently available evidence.” To our best knowledge, no other articles exist within the currently available literature that compare the costs of L-TME and TaTME head-to-head. Subsequently, there is a lack of evidence on the exact factors that lead to the reported costs difference between L-TME and TaTME that warrant further research. Furthermore, in the included article by Elbarmelgi et al., total costs of TaTME were reported. Though the article reported these costs were based on instrument costs, consumable costs and ward costs, the actual costs of these specific fields were not provided.

Considering the additional reference as proposed by the review, we believe that the article provides an insightful overview on the present roll and future prospects of TaTME, and is of added value when evaluating the comparison between L-TME and TaTME in the discussion. Therefore, we have added the following sentence including the reference to the “Discussion” section on page 18, paragraph 1, line 345, 346, 347 & 348: “As reported by Vignali et al., currently available evidence suggests short-term (post)operative and oncologic results of TaTME are comparable to L-TME. However, analyses of the costs related to the procedure are required, in particular when a two-team operation is performed.

Comment #3:

- In the section “laparoscopic versus robot-assisted TME” it has been reported that “four studies showed significantly higher total costs of R-TME compared to L-TME while one study reported no significant differences. Two studies reported higher operative costs for R-TME while the hospitalization costs were lower for R-TME. Instrumentation costs were higher in R-TME than L-TME while conversion costs were lower in R-TME than L-TME. Lastly, costs of complications and ICU stay were significantly higher in R-TME than L-TME”.

How do you explain the higher costs of complications and ICU stay in R-TME than L-TME? This topic should be enhanced.

Moreover evaluating the comparison between laparoscopic and robot-assisted TME, why don’t you mention this paper?

Milone M, Manigrasso M, Velotti N, Torino S, Vozza A, Sarnelli G, Aprea G, Maione F, Gennarelli N, Musella M, De Palma GD. Completeness of total mesorectum excision of laparoscopic versus robotic surgery: a review with a meta-analysis. Int J Colorectal Dis. 2019 Jun;34(6):983-991. doi: 10.1007/s00384-019-03307-0. Epub 2019 May 6. PMID: 31056732.

The authors would again like to sincerely appreciate this remark and the valuable additional literature. In order to explain the higher costs of complications and ICU stay in R-TME than L-TME reported by Rouanet et al., we’ve added the following sentences to the “Discussion” section on page 18 line 360: “This may be explained as this study assessed patients receiving R-TME in 2015 performed by surgeons of which some may still have been in their respective robotic learning curve, which is associated with increased morbidity. Moreover, when assessing a cohort of patients receiving R-TME in 2018, after the robotic learning curve had been achieved and the team was familiar with the ERAS procedures involved, the authors reported that costs were reduced to a level that was comparable to L-TME, with lower costs in patients without co-morbidity. “ The effects of the learning curve on the hospitalization costs are further elaborated in the “Discussion” section on page 18, paragraph 2.

Considering the additional reference as proposed by the review, we believe that the article provides an insightful overview on the comparison between R-TME and L-TME, that would add valuable information to our introduction. Therefore, we have the following sentence including the reference to the “Introduction” section on page 6, paragraph 2, line 115, 116 & 117: “Similarly, a systematic review and meta-analysis by Milone et al. reported R-TME was associated with significantly higher rates of complete TME resection, whereas L-TME had a higher incidence of nearly complete TME.”

Comment #4:

- In the section “laparoscopic versus robotic versus open TME” Ramji et al. reported significantly higher total costs of R-TME compared to L-TME and open TME, operative costs were significantly higher in R-TME compared to L-TME and open. Is there any information about specific fields such as post-discharge costs and indirect costs in current literature?

We would to thank the review for his comment. As depicted in Table 2, the article by Ramji et al. provides information on a number of cost components, including ward costs, pharmacy costs, laboratory costs, PACU costs and imaging. Overall, the higher total operative costs of R-TME are attributed to the higher operative costs in this article. The article does not contain any information on post-discharge costs and indirect costs. Furthermore, to the best of our knowledge, no studies are currently available in literature that report on the post-discharge cost and indirect costs of open TME, L-TME or R-TME. Hence, we proposed for more high quality research that also include the indirect cost in their cost analysis in lines 446, 447 & 448. Also, we have changed the following sentence in the “Discussion” section on page 22, paragraph 1, line 446, 447 & 448 from: ”In addition, these comparative studies should be performed from a societal perspective and include indirect costs.” into: In addition, these comparative studies should be performed from a societal perspective and include indirect costs, particularly as these are not yet included in currently available evidence.”

---

## [Editor Report · Decision Letter 1]

11 Jul 2023

Economic analysis of open versus laparoscopic versus robot-assisted versus transanal total mesorectal excision in rectal cancer patients: a systematic review.

PONE-D-23-09962R1

Dear Dr. Geitenbeek,

We’re pleased to inform you that your manuscript has been judged scientifically suitable for publication and will be formally accepted for publication once it meets all outstanding technical requirements.

Kind regards,

Michele Manigrasso

Academic Editor

PLOS ONE
---

## [Editor Report · Acceptance letter]

20 Jul 2023

PONE-D-23-09962R1 

Economic analysis of open versus laparoscopic versus robot-assisted versus transanal total mesorectal excision in rectal cancer patients: a systematic review. 

Dear Dr. Geitenbeek:

I'm pleased to inform you that your manuscript has been deemed suitable for publication in PLOS ONE. Congratulations! Your manuscript is now with our production department. 

Kind regards, 

on behalf of

Dr. Michele Manigrasso 

Academic Editor

PLOS ONE